# HH-NIDS: Heterogeneous Hardware-Based Network Intrusion Detection Framework for IoT Security

**Duc-Minh Ngo** [1,*] **, Dominic Lightbody** [1]**, Andriy Temko** [1]**, Cuong Pham-Quoc** [2,*]**, Ngoc-Thinh Tran** [2]**, Colin C. Murphy** [1,*] **and Emanuel Popovici** [1]

1   Electrical and Electronic Engineering, University College Cork, T12 K8AF Cork, Ireland
2   Computer Science and Engineering, Ho Chi Minh City University of Technology (HCMUT), VNU-HCM, 268 Ly Thuong Kiet St., Dist. 10, Ho Chi Minh City 740050, Vietnam
*   Correspondence: 120220051@umail.ucc.ie (D.-M.N.); cuongpham@hcmut.edu.vn (C.P.-Q.); colinmurphy@ucc.ie (C.C.M.)

**Abstract:** This study proposes a heterogeneous hardware-based framework for network intrusion detection using lightweight artificial neural network models. With the increase in the volume of exchanged data, IoT networks' security has become a crucial issue. Anomaly-based intrusion detection systems (IDS) using machine learning have recently gained increased popularity due to their generation's ability to detect unseen attacks. However, the deployment of anomaly-based AI-assisted IDS for IoT devices is computationally expensive. A high-performance and ultra-low power consumption anomaly-based IDS framework is proposed and evaluated in this paper. The framework has achieved the highest accuracy of 98.57% and 99.66% on the UNSW-NB15 and IoT-23 datasets, respectively. The inference engine on the MAX78000EVKIT AI-microcontroller is 11.3 times faster than the Intel Core i7-9750H 2.6 GHz and 21.3 times faster than NVIDIA GeForce GTX 1650 graphics cards, when the power drawn was 18mW. In addition, the pipelined design on the PYNQ-Z2 SoC FPGA board with the Xilinx Zynq xc7z020-1clg400c device is optimised to run at the on-chip frequency (100 MHz), which shows a speedup of 53.5 times compared to the MAX78000EVKIT.

**Keywords:** network security; artificial neural Networks; hardware accelerators; low power; high-performance; microcontrollers; CPU; GPU; FPGA





## 1. Introduction

In recent years, the Internet has witnessed significant growth in data volume due to the increase in data communication between Internet-connected devices worldwide. In addition, the rapid internet of things (IoT) development brings many new devices into the global Internet; this number is predicted to rise to 25 billion devices in 2030 [1]. These devices often collect users' data and store it on the network, and, most importantly, many manufacturers are approaching the IoT device market, rushing to release new products continuously. This aspect has led to poor product design with open security vulnerabilities. This phenomenon creates challenges for network administrators to protect the system from hackers, who could examine the system with malicious emails, DoS/DDoS attacks and other malicious worms or Trojans.

Intrusion detection systems (IDS) can be built based on signature-based or anomaly-based approaches [2]. The signature-based IDS scans each incoming network packet using stored rules in the system's database; thus, it can prevent known attacks with a high accuracy rate. In the signature-based IDS, the vendor has to update the signature database regularly [3]. These systems require not only additional storage resources, when new rules are added over time, but also introduce extra overhead [4] while querying the ever-growing database. Furthermore, signature-based IDS cannot detect new attacks, which can potentially crash the system. Therefore, state-of-the-art IDS are applying anomaly-based approaches [5] to be able to detect unknown attacks. In addition, anomaly-based approaches

can be implemented using lightweight mathematical models, which rely primarily on computational resources rather than massive storage, compared to signature-based techniques.

In the meantime, the development of Artificial Intelligence (AI) has dramatically impacted many areas, including IDS implementations. AI applications have become a prominent topic because this technology can train machines to mimic human cognitive functions. It has achieved success in solving many practical problems, such as speech-recognition [6], encryption [7], biosignal monitoring [8], etc. Besides, machine learning (ML) [9] is renowned as an application of AI that can improve itself based on experience. ML has shown high efficiency and applicable results in predicting attacks in anomaly-based IDS, which is one of the classification problems. Many ML classification techniques [10] such as linear classifiers, logistic regression, support vector machines (SVM), decision trees (DT) and artificial neural networks (ANN) have been used to predict the category to which the data belongs.

From the IoT network point of view, anomaly-based IDS have to be carefully designed and implemented due to the device's limitations. On the one hand, IoT devices come from different manufacturers, requiring additional hardware configurations and connections. In other words, there has yet to be a consensus regarding protocols and standards between IoT devices; as a result, deploying and establishing communications for these devices is time-consuming. On the other hand, many devices need continuous power to function correctly; integrating ML applications into IoT devices has to consider both power consumption and computational capability. Additionally, anomaly-based IDS run real-time applications that heavily depend on hardware platforms. Thus, it is important to choose suitable hardware for prototyping and deploying IoT applications.

Hardware accelerators such as Graphics Processing Units (GPUs), low-power microprocessors and Field-Programmable Gate Arrays (FPGAs) have a high potential use for IoT devices running ML applications. This paper introduces an anomaly-based IDS framework on heterogeneous hardware for IoT devices. The framework aims to generate high-efficiency and low-power consumption ANN models for deploying on hardware accelerators.

### 1.1. Contribution

This research is extended from our existing FPGA Hardware Acceleration Framework [11]. This paper introduces HH-NIDS—a heterogeneous hardware-based network intrusion detection framework for IoT security. HH-NIDS applies anomaly-based IDS approaches for IoT devices using hardware accelerators. The first prototype system utilises the supervised-learning method on the IoT-23 [12] and UNSW-NB15 [13] datasets to generate lightweight ANN models. We implement the proposed framework on a heterogeneous platform, including a microcontroller, an SoC FPGA and GPU platforms under the handling of host processors. The training phase is conducted on GPU, while the inference phase is accelerated on two hardware platforms:

- The MAX78000 AI microcontroller with ultra-low power consumption [14].
- The SoC FPGA is implemented in high-level synthesis (HLS) and Verilog approaches. The Verilog implementation is customised in a pipeline fashion, which has achieved high-throughput and low latency in processing network packets.

Finally, the first prototype system is deployed on the MAX78000 evaluation kit [15] (MAX78000EVKIT) and PYNQ-Z2 SoC FPGA board [16]. The implemented systems are evaluated using a number of testing scenarios on the two public datasets: IoT-23 and UNSW-NB15. Hardware resource usage and power consumption are also discussed and compared to relevant works.

### 1.2. Organisation

The rest of this work is organised as follows. Section 2 introduces modern anomaly-based IDS for IoT networks. In Section 3, we present the materials and methods used here for framework design. Section 4 shows our first prototype system based on the

proposed hardware-based architecture. We evaluate and analyse our implemented system in Section 5. Next, we discuss the advantages and disadvantages of HH-NIDS in Section 6. Finally, the conclusions and future work suggestions are stated in Section 7.

## 2. Related Works

Machine learning and deep learning (DL) have been widely adopted in anomaly-based IDS to detect unseen attacks [17–19]. The authors in [20,21] have experimented with various machine learning algorithms on an IoT-based open-source dataset. They have achieved up to 99% accuracy by applying the random forest (RF) classifier; however, the dataset used in this research was small (357,952 records) compared to other practical datasets. RF has also been claimed, in [22], as the best algorithm for detecting anomalies. Meanwhile, DL has shown better performance when dealing with large datasets. For instance, researchers in [23] have trained deep neuron network (DNN) models on six different datasets with the best training accuracy in the range 95% to 99% on the KDDCup-99 and NSL-KDD datasets. On the same datasets, the system in [24] has achieved similar results (up to 99% detection rate) by using a deep network model with an automatic feature extraction method. Even though ML and DL are prominent techniques for anomaly-based IDS with high detection rates, the trained DL models usually require heavy computational resources, which are limited on standard IoT devices.

Table 1 shows a summary of anomaly-based IDS for IoT networks. The paper [25] has introduced a federated self-learning anomaly detection in IoT networks on a self-generated dataset. Similarly, authors in [26] have proposed a decentralised federated-learning approach with an ensembler, which combines long short-term memory (LSTM) and gated recurrent units to enable anomaly detection. Researchers in [27] have tested various ML algorithms on a generated dataset, called MQTT, for IoT network attack detection and achieved an accuracy of 98%. DL algorithms have also been used to generate detection models from various IoT datasets. For instance, authors in [28] have built deep belief network models from the CICIDS 2017 dataset to classify normal records from six different attack types, with an average accuracy of 97.46%. In [29], the Yahoo webscope s5 dataset has been used as input for convolutional neuron network (CNN) and recurrent autoencoder algorithms. Authors in [30] have generated lightweight detection models based on a deep autoencoder from the Bot-IoT dataset. The best setup has reached 97.61% F1-score; however, the hardware platform for experimenting with the system is not mentioned. The self-generated dataset is collected in [31]; the data are fed into a graph neural network, which has produced up to 97% accuracy in the literature.

**Table 1.** Summary of anomaly-based IDS for IoT networks.

| Work | Publication Year | Dataset | Platform | Accuracy |
|------|------------------|---------|----------|----------|
| [25] | 2019 | Self-generated | Redeon Rx 460 GPU | 95.6% |
| [26] | 2020 | Modbus network | Lambda GPU | 99.5% |
| [27] | 2020 | MQTTset | Core Intel i7 dual, 16 GB of RAM | 98.0% |
| [28] | 2020 | CICIDS 2017 | Core Intel i7, 16 GB of RAM | 99.4% |
| [29] | 2021 | Yahoo Webscope S5 | Google Colab | 99.6% |
| [30] | 2021 | Bot-IoT | Not mentioned | 99.0% |
| [31] | 2021 | Self-generated | Not mentioned | 97.0% |

The IoT-23 dataset has been trained with supervised-learning methods. Table 2 presents the summary of anomaly-based IDS on the IoT-23 dataset. The authors in [32,33] have applied an ensemble strategy, which combines DNN and LSTM. The paper [34] has proposed a universal feature set to detect botnet attacks on the IoT-23 dataset. However, the proposed feature extraction module is tool-based, which can cause dependency and add

extra overhead [35]. Malware and botnet analysis on the IoT-23 dataset has been examined in [36,37]. Authors in [38] have reported 99.9% accuracy by applying hyper-parameter optimisation on the deep ensemble model (CNN and RF). Meanwhile, ML algorithms have been used in [39,40] to generate lightweight detection models, resulting in 99.62% and 99.98% accuracy, respectively. However, there are limitations in current works. On the one hand, most of the anomaly-based IDS for IoT networks are designed at the system's high-level view that is usually deployable at the IoT gateway level. Anomaly behaviours can be detected earlier at the edge devices. On the other hand, they have focused on software-based metrics only, while hardware-based metrics, such as power consumption, performance and network throughput, are not mentioned.

**Table 2.** Summary of anomaly-based IDS on the IoT-23 dataset.

| Work | Publication Year | Dataset | Platform | Accuracy |
|---|---|---|---|---|
| [32] | 2020 | IoT-23, LITNET-2020 and NetML-2020 | Not mentioned | 99.0% |
| [33] | 2022 | NSL-KDD, BoT-IoT, IoT-NI, IoT-23, MQTT and IoT-DS2 | Not mentioned | 99.8% |
| [34] | 2020 | CICIDS2017, CTU-13 and IoT-23 | Not mentioned | 99.9% |
| [36] | 2020 | IoT-23 | GTX 1060, 6 GB of RAM, | 99.5% |
| [37] | 2020 | IoT-23 and Self-generated | Microsoft Azure | 99.8% |
| [38] | 2022 | IoT-23 | Intel Core i7-9750H, 32 GB of RAM | 99.9% |
| [39] | 2022 | TON-IoT, IoT-23 and IoT-ID | Not mentioned | 99.6% |
| [40] | 2022 | NSL-KDD, IoT-23, BoT-IoT and Edge-IIoT | GPU GeForce MX330, 8 GB of RAM | 99.9% |

Hardware accelerators, including specialised AI microcontrollers and SoC FPGA, are promising platforms for securing IoT devices [41,42]. IoT security monitoring systems [43,44] have implemented accelerating semantic caching on FPGA. Researchers in [45] have introduced a mapping methodology to bring neuron network-based models to hardware. The proposed method has achieved a latency of 210 ns on a Xilinx Zynq UltraScale+ MPSoC FPGA running mapped LSTM models. FPGA acceleration for time series similarity prediction [46] has been proposed for IoT edge devices. The implemented system can reach a throughput of 453.5G operations per second with a 10.7x faster inference rate on a Xilinx Ultra96-V2 FPGA compared to the Raspberry Pi. Lightweight ANN models for intrusion detection systems are proposed on an SoC FPGA platform in [47]. They have achieved a speedup of 161.7 times faster than software-based methods on the NSL-KDD dataset. Overall, anomaly-based IDS for IoT devices on hardware accelerators is a promising approach. This paper introduces an ultra-low power consumption and low-latency heterogeneous hardware-based IDS framework. The prototype system is tested with the IoT-23 and UNSW-NB15 datasets on the MAX78000 microcontroller and SoC FPGA platforms.

## 3. Materials and Methods

### 3.1. Dataset

The IoT-23 dataset [12] has been created as part of the Avast AIC laboratory. This dataset was captured in twenty-three scenarios from 2018 to 2019 and was published in

January 2020. It contains network packets from IoT devices to offer a comprehensive dataset. The network packets are extracted to produce labelled malware-infected and benign records. Table 3 illustrates the total of 325,307,990 labelled records belonging to sixteen categories. Each record has eighteen features, which can be divided into four groups: static, basic, content and time features. Static features can be extracted from the packet header fields. There are five static features in the IoT-23 dataset: IP source, IP destination, source port, destination port and protocol. Other features are calculated from packets in the same flow (or in a time window).

**Table 3.** The IoT-23 dataset record distribution.

| # | Label | Number of Records |
|---|---|---|
| 1 | C&C-Mirai | 2 |
| 2 | PartOfAHorizontalPortScan-Attack | 5 |
| 3 | C&C-HeartBeat-FileDowbload | 11 |
| 4 | FileDownload | 18 |
| 5 | C&C-Torii | 30 |
| 6 | C&C-FileDownload | 53 |
| 7 | C&C-HeartBeat-Attack | 834 |
| 8 | C&C-PartOfAHorizontalPortScan | 888 |
| 9 | Attack | 9398 |
| 10 | C&C | 21,995 |
| 11 | C&C-HeartBeat | 33,673 |
| 12 | DDoS | 19,538,713 |
| 13 | Benign | 30,858,735 |
| 14 | Okiru-Attack | 13,609,470 |
| 15 | Okiru | 47,381,241 |
| 16 | PartOfAHorizontalPortScan | 213,852,924 |

The UNSW-NB15 dataset [13] is a hybrid of real modern normal activities and synthetic contemporary attack behaviours. This dataset was generated by the IXIA PerfectStorm tool in the Cyber Range Lab of UNSW Canberra. There are 100 GB of raw network packets (pcap files), which are captured using the tcpdump tool. Table 4 shows the UNSW-NB15 dataset record distribution for training and inference purposes with a total of 2,540,044 labelled records belonging to ten categories. The dataset has nine types of attacks: Worms, Shellcode, Backdoors, Analysis, Reconnaissance, DoS, Fuzzers, Exploits and Generic.

**Table 4.** USNW-NB15 dataset record distribution.

| # | Label | Number of Records |
|---|---|---|
| 1 | Worms | 174 |
| 2 | Shellcode | 1511 |
| 3 | Backdoors | 2329 |
| 4 | Analysis | 2677 |
| 5 | Reconnaissance | 13,987 |
| 6 | DoS | 16,353 |
| 7 | Fuzzers | 24,246 |
| 8 | Exploits | 44,525 |
| 9 | Generic | 215,481 |
| 10 | Benign | 2,218,761 |

### 3.2. Evaluation Metrics

The confusion matrix is widely used to evaluate classification performance. It is a table that shows where the true values belong. Evaluation metrics, including accuracy, precision, recall and F1-score are computed from this confusion matrix. The definition of

True Positive (TP), False Positive (FP), False Negative (FN) and True Negative (TN) for multiple classes can be provided from the confusion matrix [48].

- Accuracy is one of the popular metrics for evaluating classification models. Equation (1) depicts the single-class accuracy measurement.

$$Accuracy = \frac{TP + TN}{TP + TN + FP + FN} \tag{1}$$

- Precision is the positive predictive value. It is a proportion of true positives that the model claims compared to the total number of positives it claims. The single class precision value is given in Equation (2):

$$Precision = \frac{TP}{TP + FP} \tag{2}$$

- Recall is defined as the actual positive rate. This number refers to the positives which the model states, compared to the total positives in the data. The single-class recall value is given in Equation (3):

$$Recall = \frac{TP}{TP + FN} \tag{3}$$

- F1-score represents the model's performance. It can be referred to the weighted average of the precision and recall values. The single class F1-score value is given in Equation (4):

$$\text{F1-score} = \frac{2 * Precision * Recall}{Precision + Recall} \tag{4}$$

### 3.3. HH-NIDS Framework

Figure 1 illustrates the proposed HH-NIDS framework architecture. The data processing flow is described in three phases, namely *Data Pre-processing*, *Model Generation*, and *Hardware-based Inference*.

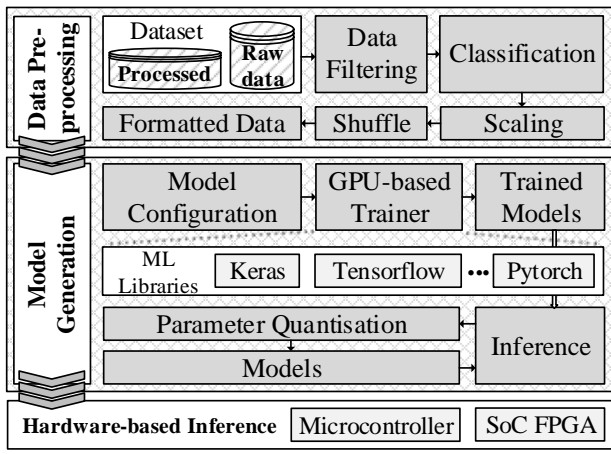

**Figure 1.** The proposed HH-NIDS framework architecture.

Data Pre-processing is the first phase in the HH-NIDS framework to generate data for training and inference processes. HH-NIDS applies a supervised learning neural network approach for detecting network attacks; thus, the chosen dataset should contain labelled records. The raw data (network packets) can be used in the inference phase for demonstrating the real-time network classification process. Besides the Dataset block, there are five main functional blocks in this pre-processing phase:

- *The Data Filtering block*: scans the processed dataset (labelled records) and buffers all of the IPv4 records, which make up the majority of the dataset, as input data for the next blocks. In other words, this block creates a custom filter for selecting input records based on the administrators' defined rules.
- *The Classification block*: examines the labelled data, and then collects records that have the same label to a new file. The number of new files is equal to the number of dataset labels/classes (benign and attacks).
- *The Scaling block*: reads the created data files and transforms the record's features, which are initially in alphanumeric or numeric forms, to float values (between 0 and 1) using min-max normalisation.
- *The Shuffle block*: rearranges records according to different training schemes. In addition, data balancing can be applied in this block for better data distribution. Although the network data can be provided with time-based information, this block shuffles these records for the neuron network models to learn static information in the network packets.
- *The Formatted Data block*: holds shuffled data files from the previous block; the data are ready to be used in the training and inference processes.

Model generation: this phase includes training, inference and quantisation processes. In the beginning, the training process generates NN models that use GPU-based approaches; then, these models are quantised to fit in accelerator platforms in the hardware-based inference phase. The training process can be described using three blocks:

- *The Model Configuration block*: has two main functionalities, which are: defining training settings and selecting a custom input feature set from each processed record for training. The output data will be forwarded to the *GPU-based Trainer block* for training.
- *The GPU-based Trainer block*: follows the ML library architectures (Keras, Tensorflow, Pytorch, etc.) to train NN models. The HH-NIDS framework proposes GPU-based approaches for better training performance compared to CPU-based approaches.
- *The Trained Models block*: stores trained models from the *GPU-based Trainer block*. These models will be used in the next inference phase.

Next, the trained models will be evaluated with test data. The administrators can monitor the validation results and select dedicated models for quantisation processes. Figure 2 illustrates the HH-NIDS framework's process flow from training to hardware implementations.

- *The Inference block*: represents the software-based inference process. This block uses the trained models on the allocated data files to evaluate how well the trained models react to unseen data. The chosen models then need to be quantised and tested before being deployed into hardware accelerator platforms.
- *The Parameter Quantisation block*: transforms the model's parameters to the supported bit width to fit into the hardware accelerator platforms.
- *The Models block*: contains quantised models, which will be evaluated again by *the Inference block*.

Hardware-based inference: this phase deploys quantised NN models into hardware accelerator platforms for inference. The HH-NIDS framework proposes using two different platforms: namely, a microcontroller and an SoC FPGA. The implemented hardware architecture is discussed in detail in Section 4.

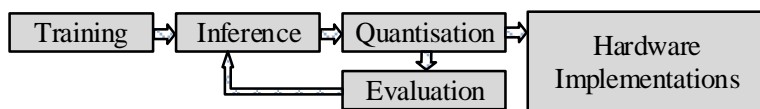

**Figure 2.** The HH-NIDS framework's processing flow from training to hardware implementations.

## 4. Implementation

In this section, we introduce HH-NIDS's prototype system implementation. The prototype system uses heterogeneous hardware, including GPU, Microcontroller, SoC FPGA and host processors for experimenting with the IoT-23 and UNSW-NB15 datasets. The implementation is explained as follows.

### 4.1. Data Pre-Processing

The HH-NIDS utilises static features for training the NN models. Table 5 shows the static features definitions from the two datasets; the source/destination IP address is represented as four 8-bit inputs, while other fields are normalised/scaled to fit into the neurons' inputs.

**Table 5.** Flow Features from the IoT-23 and UNSW-NB15 datasets.

| # | IoT-23 | UNSW-NB15 | Inputs | Description |
|---|--------|-----------|--------|-------------|
| 1 | id_orig.h | srcip | 4 | Source IP address |
| 2 | id_orig.p | sport | 1 | Source port number |
| 3 | id_resp.h | dstip | 4 | Destination IP address |
| 4 | id_resp.p | dsport | 1 | Destination port number |
| 5 | proto | proto | 1 | Transaction protocol |

### 4.2. Model Generation

As described in the previous section, the pre-processing data phase extracts eleven flow features as inputs for our NN models. The hidden layer in a NN model is configured with 32 or 48 neurons, each followed by a hardware-efficient ReLU (Rectified Linear Unit) activation function; the softmax activation function is used in the output layer. The processed data are allocated at the rate of 50% for training, 25% for validation, and 25% for inference. The NN models were trained using the cross-entropy loss function and Adam optimiser with a batch size of 512, and the learning rate is set equal to 0.0005, 0.001, or 0.01.

### 4.3. Microcontroller

The inference phase has been deployed on the Maxim MAX78000EVKIT, which has an artificial intelligence microcontroller with ultralow-power neural network acceleration processors. Maxim provides sample projects, which generate NN models using the PyTorch library. The NN processors on the MAX78000EVKIT can only support a maximum of 8-bit registers for storing parameters (weights and bias). As a result, the training process is configured as quantisation-aware.

### 4.4. SoC FPGA

Figure 3 illustrates the inference phase block diagram on the Zynq-7000 SoC architecture, which includes two main domains: namely, the Processing System (PS) and the Programmable Logic (PL/FPGA). The PS has basic modules for a computer system: Multiplexer Input/Output (MIO), I/O Peripherals, Memory (MEM), Interconnects and an Application Processor Unit (APU). The SoC FPGA board is booted from an SD card, and the programmed application is loaded into the APU. Here, network packets will be sent to the APU for extracting input features in the *Feature Extraction* module. Then, the extracted features are grouped together in the *Buffer Initialization* module, before being sent to the PL. This buffer helps reduce overhead in data exchanges between the PS and the PL. Processed results from the PL are delivered back to the *Monitoring* module for analysis. The PL domain has five main blocks, which are:

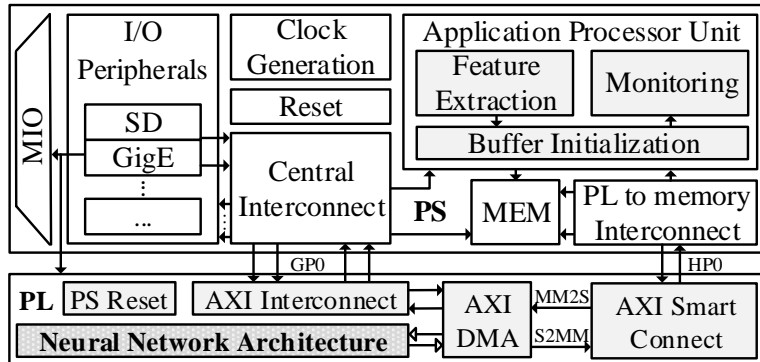

**Figure 3.** Neural network inference acceleration on Zynq-7000 SoC architecture.

- *The PS Reset* block: receives the reset control signals from the PS to reset the PL to the initial state.
- *The AXI Interconnect* block: assigns user's input values to registers in PL through General-Purpose Ports (GP0), which is configured as a 32-bit AXI-lite interface. There are two registers to be configured in the *NN* block: namely, the *RESET* register for resetting the module and the *NUM_OF_INPUT* register to indicate the number of records to be sent from PS each time.
- *AXI Smart Connect* block: receives grouped input features, which are buffered in the PS, using High-Performance Ports (HP0). Then, data are sent to the *AXI DMA* block by a Memory-Mapped to Stream (MM2S) channel. In addition, the scanned data are transferred back from *AXI DMA* to this block by a Stream to Memory-Mapped (S2MM) channel. These data are intrusion detection results that are ready to be sent back to PS for analysis.
- *The AXI DMA* block: this is connected to the *NN* block through an AXI Stream interface for sending/receiving data to/from the *NN* block.
- *The Neuron Network Architecture* block: this contains the NN model on FPGA. This block is implemented in two approaches: namely, high-level synthesis (HLS) and Verilog implementation.

### 4.4.1. HLS Approach

The HLS approach builds the NN block from a C implementation. Figure 4 describes the NN block's architecture on FPGA. The trained parameters are loaded into BlockRAMs memory, and the input for neurons is extracted from the input stream by the *Pre-processor* module. The hidden and output layers are configured using a pipeline strategy and resource-sharing techniques in which there are only eight neurons in each layer. Additionally, DSP48 blocks are used to implement both MUL and MAC operands. Finally, the *Post-processor* module produces the output stream, which connects to the *AXI DMA* block.

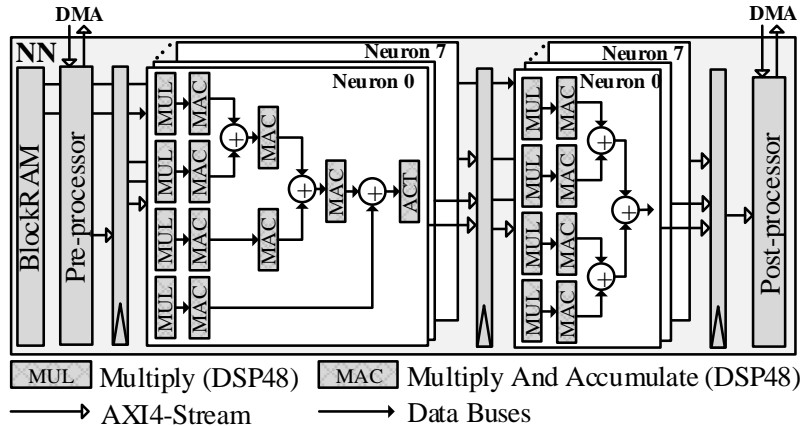

**Figure 4.** The generated FPGA block architecture using the HLS implementation approach.

The NN block on FPGA is implemented by Vivado HLS 2018.2. Parameter quantisation and HLS Pragmas techniques are taken into account for implementation, in particular:

- *Parameter Quantisation*: this transforms each trained parameter of an NN model to integer values. For instance, for a layer with weight values in the range of $[-w, w)$, a quantised value, $w_{quantised}$, from $w_i$, is calculated by

$$w_{quantised} = \lfloor 2^n \frac{w_i}{w} \rfloor \tag{5}$$

where *n* is the number of bits for storing a weight value in the FPGA. This number is set to 32 in the case of HLS implementation.

- *HLS Pragmas* are applied for optimising our implementation in C++ into the RTL. For instance, *HLS Dataflow* is used for pipelining calculations between layers, *HLS Pipeline* and *HLS Unroll* are used for parallel computation of neurons in each layer.

4.4.2. Verilog Implementation Approach

This approach uses a hardware description language to implement the *Neuron Network Architecture* block. Figure 5 shows the prototype implementation on an FPGA-based architecture. The data flow is similar to the *HLS implementation approach*; however, the number of neurons in the hidden and output layers is maximised for the designed neuron network architecture (11 inputs, 32 hidden neurons, 16 output neurons).

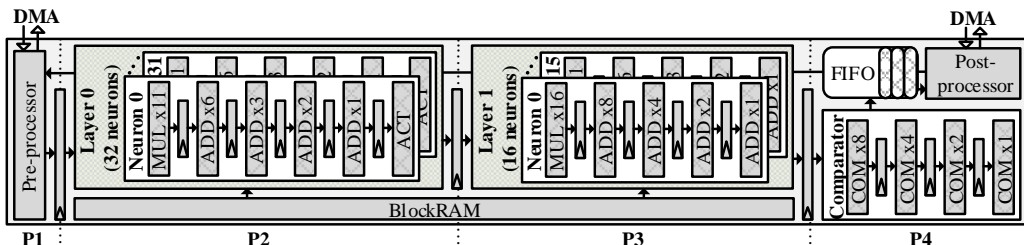

**Figure 5.** Pipeline implementation of the *Neuron Network Architecture* block on an FPGA-based architecture includes four phases: Input buffering (P1), Layer_0 calculation (P2), Layer_1 calculation (P3), and Output buffering (P4).

By using Verilog, the processing performance is optimised manually with a pipeline strategy, in which buffers are placed between each calculation step. The pipelined NN architecture could be described as having four phases:

- *Input buffering* (P1): contains *the Pre-processor* block to receive the eleven input features from DMA through the AXI4-Stream interface.

- *Layer_0 calculation* (P2): has 32 neurons running simultaneously. These neurons have the same architecture, which has six calculation stages: multiplication, four additions, and an activation. Figure 6 illustrates a single neuron architecture on FPGA. From the P1 phase, the 11-input features are fed into the corresponding eleven 2-input multipliers (*MULx 11* block). Next, these results are accumulated by the next four addition stages, represented as *ADD x6*, *ADD x3*, *ADD x2*, and *ADD x1* blocks. The final summed result is passed to the *ACT* block, which is implemented as a hardware-efficient ReLU (Rectified Linear Unit) activation function.

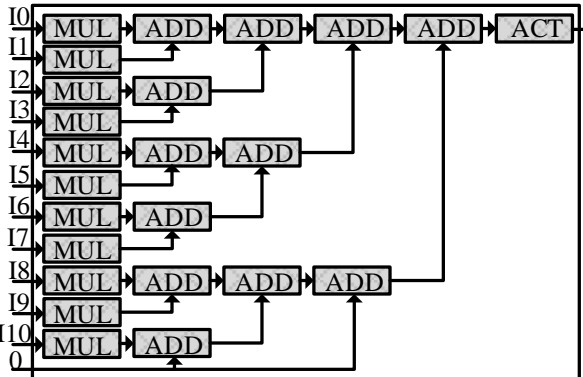

**Figure 6.** The *MULx11* block calculation architecture.

- *Layer_1 calculation* (P3): receives buffered results from the P2 phase; then, these data are distributed to sixteen neurons in this block. Each neuron shares the same calculation architecture with Layer_0 neurons; except that each neuron in Layer_1 has five calculation stages: a multiplication (*MUL x16* block) and four additions (*ADD x8*, *ADD x4*, *ADD x2*, and *ADD x1* blocks). The final results are buffered and will be read in the next clock cycle.
- *Output buffering* (P4): includes the *Comparator*, the *FIFO buffer*, and the *Post-processor* blocks. The *Comparator* block returns the maximum index from sixteen inputs, corresponding to sixteen output neurons in the P3 phase. These indices are stored in a FIFO (First in, first out) buffer. The *Post-processor* block reads data from the buffer and sends it to DMA; this block also sends a signal to the *Pre-processor* block to indicate if the buffer can store all the results.

## 5. Results

In this section, the prototype HH-NIDS implementation is evaluated. Firstly, we discuss the hardware resource usage on two platforms: microcontroller (MAX78000EVKIT) and SoC FPGA (PYNQ-Z2). Secondly, the accuracy results are discussed and compared to relevant works. Finally, the hardware performance is measured.

### 5.1. Implemented Result

On the MAX78000 microcontroller, the NN models have two parameter layers with a total of 1360 parameters, which corresponds to 1360 bytes. With respect to the total hardware resources on the board, 1360 bytes out of 442,368 bytes (0.3%) weight memory were used. Six bytes out of 2048 bytes of total bias memory were utilised (the bias was not used). The records were sent, one after another, repeatedly. Utilising the power monitor built into the MAX78000EVKIT, the power drawn was between 17 mW (loading weights etc.) and 18mW (when the NN model is inferring).

As introduced in Section 4, our NN models on FPGA are implemented using two different approaches: namely, HLS and Verilog. These implementations on Vivado version 2018.2 were deployed on a PYNQ-Z2 board, which includes Dual ARM Cortex-A9 MPCore and a Xilinx Zynq xc7z020-1clg400c SoC (containing a total of 53,200 LUTs, 106,400 FFs, 140 BRAMs, and 220 DSP slices).

Table 6 shows the hardware resource usage, timing and power consumption estimation of the implemented NN for the IoT-23 dataset. In terms of FPGA resource usage, the HLS implementation approach consumed 24.60% LUTs, 15.25% FFs; while the Verilog implementation approach used nearly double these resources, which are 41.97% LUTs and 26.60% FFs on the PYNQ-Z2 board. Although the Verilog implementation approach used fewer BRAMs (14.29% compared to 30.36%), this approach required 220 DSP slices, which is 100% of the on-chip DSP slices on the PYNQ-Z2 board. The maximum frequency is approximately 101MHz in both implemented strategies. The power synthesised in the Verilog implementation is 0.3W less than the HLS implementation (1.53 W compared to 1.83 W).

**Table 6.** The IoT-23 dataset implemented results on PYNQ-Z2 SoC FPGA.

| Hardware Resources Usage | | | | |
|---|---|---|---|---|
| **Approach** | **Resources** | **Utilisation** | **Available** | **Utilisation (%)** |
| HLS | LUT | 13,089 | 53,200 | 24.60 |
| | LUTRAM | 242 | 17,400 | 1.39 |
| | FF | 16,224 | 106,400 | 15.25 |
| | BRAM | 42.5 | 140 | 30.36 |
| | DSP | 152 | 220 | 69.09 |
| Verilog | LUT | 22,329 | 53,200 | 41.97 |
| | LUTRAM | 1420 | 17,400 | 8.16 |
| | FF | 28,304 | 106,400 | 26.60 |
| | BRAM | 20 | 140 | 14.29 |
| | DSP | 220 | 220 | 100 |
| Timing and Power | | | | |
| | **F_max** | **Static** | **Dynamic** | **Total** |
| HLS | 102.1 MHZ | 0.15 W | 1.68 W | 1.83 W |
| Verilog | 101.2 MHZ | 0.14 W | 1.39 W | 1.53 W |

Similarly, Table 7 represents the hardware resource usage, timing and power consumption estimation of the implemented NN models on the UNSW-NB15 dataset. The used LUTs and FFs are 22.15% and 15.26% in the HLS implementation approach; these numbers are 52.64% and 31.93% in the Verilog implementation approach, respectively. The BRAMs and DSP resources used are approximately equal to those of the IoT-23 dataset implementations. The maximum frequency is approximately 102.4 MHz in both approaches, while the power consumption in the Verilog implementation is 0.23 W less than in the HLS implementation approach.

### 5.2. Accuracy

The training, validation and inference accuracy will be discussed in this section. There are two training schemes corresponding to the targeted hardware: MAX78000EVKIT and PYNQ-Z2. The training processes were run in each scenario with three different learning rates (LR): 0.0005, 0.001, and 0.01 on the UNSW-NB15 and IoT-23 datasets.

Figure 7 depicts the training and validation accuracy in the first scheme, which was aimed at the MAX78000EVKIT. In detail, the line chart in Figure 7a shows trained results on the UNSW-NB15 dataset. The highest LR (0.01) returned the best score at 97.96% accuracy (epoch 8). However, this number dropped significantly, to approximately 96.82%, in the rest of the training process, after the quantisation-aware training was activated. The training processes with lower LR reported better results, which drew increasing trends in validation accuracy, from 97.12% to the peak at 97.84% from epoch 9 to epoch 40 (with LR at 0.001).

**Table 7.** The UNSW-NB15 dataset implemented results on PYNQ-Z2 SoC FPGA.

| Hardware Resources Usage | | | | |
|---|---|---|---|---|
| **Approach** | **Resources** | **Utilisation** | **Available** | **Utilisation (%)** |
| HLS | LUT | 11,783 | 53,200 | 22.15 |
| | LUTRAM | 242 | 17,400 | 1.39 |
| | FF | 16,236 | 106,400 | 15.26 |
| | BRAM | 44 | 140 | 31.43 |
| | DSP | 152 | 220 | 69.09 |
| Verilog | LUT | 28,004 | 53,200 | 52.64 |
| | LUTRAM | 1412 | 17,400 | 8.11 |
| | FF | 33,974 | 106,400 | 31.93 |
| | BRAM | 20 | 140 | 14.29 |
| | DSP | 219 | 220 | 99.55 |
| **Timing and Power** | | | | |
| | **F_max** | **Static** | **Dynamic** | **Total** |
| HLS | 102.5 MHZ | 0.15 W | 1.64 W | 1.79 W |
| Verilog | 102.4 MHZ | 0.14 W | 1.42 W | 1.56 W |

Figure 7b illustrates the training and validation accuracy on the IoT-23 dataset. The accuracy can reach up to 99.98% before dropping dramatically to 89.38% (0.001 LR) and 74.94% (0.01 LR) due to the activation of quantisation-aware training. The 0.0005 and 0.001 LR training processes showed a slight improvement over time, with training accuracy around 92.85%. The lower LR values (0.0005 and 0.001) produced better models in the quantisation-aware training processes. The accuracy fluctuated if LR was 0.01 on both datasets.

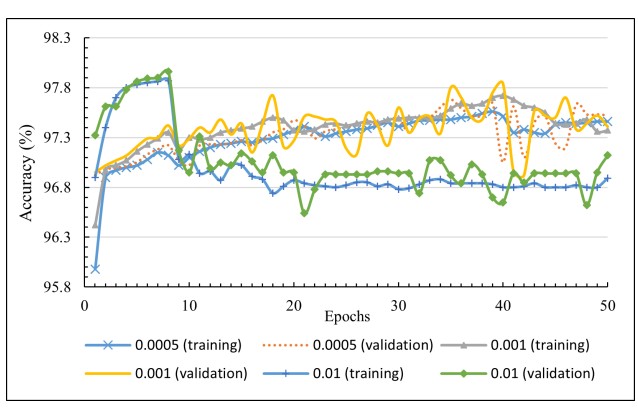

(**a**) Results on UNSW-NB15 dataset.

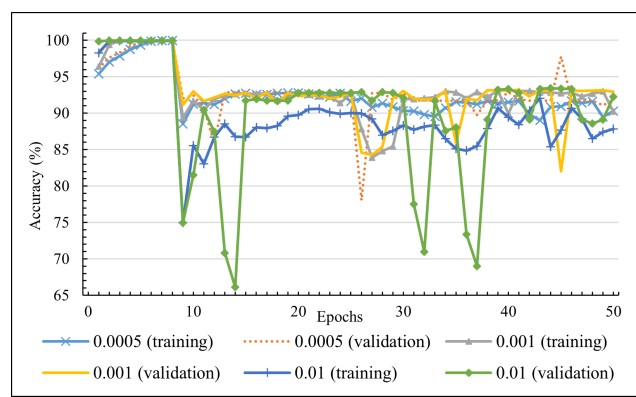

(**b**) Results on IoT-23 (2) dataset.

**Figure 7.** The trained results in fifty epochs with three different learning rates: 0.0005, 0.001, and 0.01 for the MAX78000 microcontroller.

In the second scheme, the training processes were aimed at the PYNQ-Z2 board. Figure 8 represents two charts showing accuracy results on the two datasets. The line chart in Figure 8a shows trained results on the UNSW-NB15 dataset. The 0.01 LR reported 96.84% accuracy in training and validation at epoch one, before increasing to 97.71% in the subsequent seven epochs; this number fluctuated slightly in the rest of the training process. The 0.001 LR also reported the same results, with the best validation accuracy being 97.70% from epoch 40, while the smallest LR (0.0005) can only reach 97.15% accuracy at epoch 50. The second line chart, which are trained results on the IoT-23 dataset, shows similar

training patterns to the first line chart. In detail, the higher LR (0.001 and 0.01) reported the highest validation accuracy at around 99.71%, while the 0.0005 LR training process achieved the best accuracy of 96.37% at epoch 47.

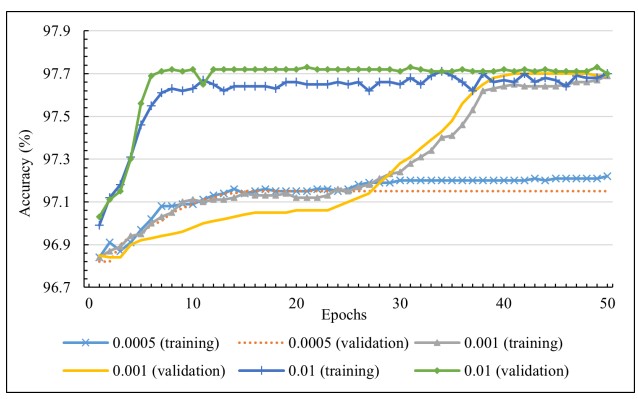

(**a**) Results on UNSW-NB15 dataset.

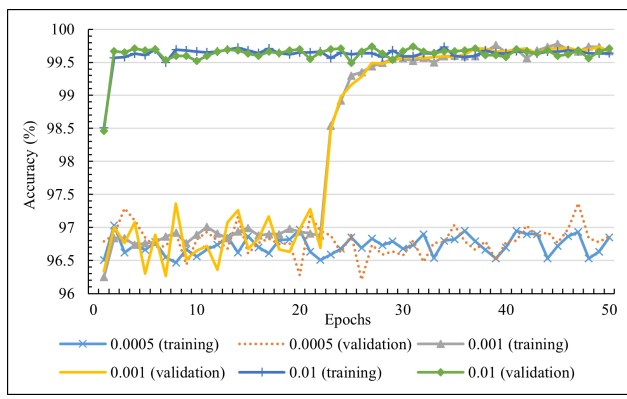

(**b**) Results on IoT-23 dataset.

**Figure 8.** The trained results in fifty epochs with three different learning rates: 0.0005, 0.001, and 0.01 for the PYNQ-Z2 SoC FPGA.

The four best models, in terms of validation accuracy, were collected from the trained results above. Evaluation metrics were calculated, including accuracy, precision, recall and F1-score on test data. Table 8 shows inference results on the MAX78000EVKIT on two models, which are denoted as (1) and (2). Model (1) on the UNSW-NB15 dataset achieves 98.57% accuracy and a 93.47% F1-score. The precision is 6.98% lower than the recall value (90.11% compared to 97.09%), which means that the model is more sensitive to attack records than normal records. In addition, model (2) on the IoT-23 dataset reports 92.69% accuracy and, 95.13% F1-score.

**Table 8.** The inference results (in %) for the two trained models on the MAX78000EVKIT.

| Dataset | Accuracy | Precision | Recall | F1-Score |
|---|---|---|---|---|
| UNSW-NB15 (1) | 98.57 | 90.11 | 97.09 | 93.47 |
| IoT-23 (2) | 92.69 | 95.04 | 95.22 | 95.13 |

Accordingly, Table 9 represents inference results on the PYNQ-Z2 board from two models, which are denoted as (3) and (4). Model (3) reproduces similar results to model (1) on the UNSW-NB15 dataset, with 98.43% accuracy and 92.71% F1-score. The highest accuracy and F1-score (99.66% and 99.81%) are achieved by the model (4) on the IoT-23 dataset.

**Table 9.** The inference results (in %) for the two trained models on the PYNQ-Z2 board.

| Dataset | Accuracy | Precision | Recall | F1-Score |
|---|---|---|---|---|
| UNSW-NB15 (3) | 98.43 | 87.95 | 98.01 | 92.71 |
| IoT-23 (4) | 99.66 | 99.97 | 99.65 | 99.81 |

We have compared both our trained NN model on the IoT-23 dataset to related works in Table 10. The authors in [32] have achieved 99.70% accuracy by using DNN. They have improved this result to 99.99% by applying LSTM in [49]; however, F1-score is not mentioned in this research. Authors in [22,34] have reported RF as the best algorithm for detecting attacks with 100% and 99.50% accuracy, respectively. Authors in [37] have achieved accuracy with only 0.14% greater than our NN model. However, their experiments have not applied to the full IoT-23 dataset [34,37].

**Table 10.** Result comparison to related works in the IoT-23 dataset.

| System | Model | Accuracy (%) | F1-Score (%) |
|--------|-------|--------------|--------------|
| [32] | DNN | 99.70 | 87.00 |
| [49] | LSTM | 99.99 | N/A |
| [34] | RF | 100.00 | 100.00 |
| [37] | NN | 99.80 | 99.85 |
| [22] | RF | 99.50 | 99.00 |
| HH-NIDS | NN | 99.66 | 99.81 |

*5.3. Performance*

Inference time is one of the most critical metrics for evaluating the HH-NIDS framework. In this section, performance measurements will be performed on the four hardware platforms, including Intel® Core™ i7-9750H 2.6 GHz CPU, NVIDIA GeForce GTX 1650 GPU, MAX78000EVKIT and PYNQ-Z2 SoC FPGA boards.

5.3.1. Inference Time

The first test measured pre-process time (allocates input buffers), inference time, and post-process time (for receiving results) of each implementation when only one input is sent. Figure 9 illustrates the five implementations' results. The bar chart shows that the inference time on the MAX78000EVKIT is only 15 µs, which is 11.3 and 21.3 times faster than CPU and GPU approaches, respectively. This number outperformed all other platforms in inference time; however, the pre-process and post-process times are not measured due to unsupported library issues. Besides, CPU and GPU implementations report the lowest pre-process and post-process time; these numbers are approximately 0.02 ms, thanks to high-frequency host processors. The inference time on CPU is 0.17 ms, while these times are 0.32 ms and 0.43 ms on GPU and FPGA, respectively. The FPGA implementations (HLS and Verilog) have higher inference times (between 0.45 ms and 0.46 ms) because of the overhead in data transfer time from the PS to the PL.

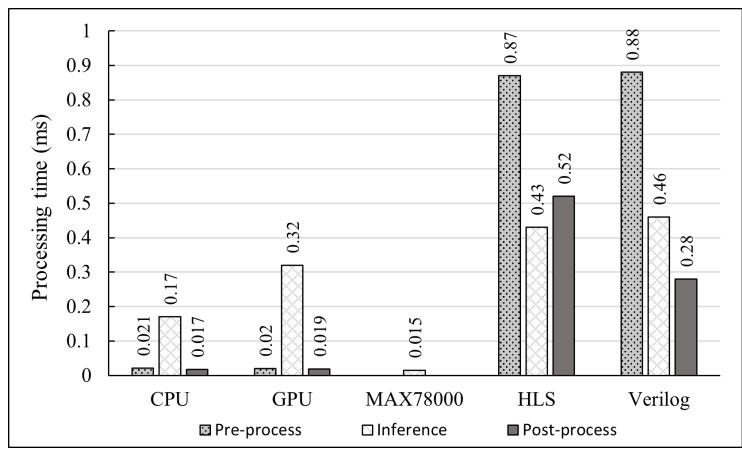

**Figure 9.** The pre-process, inference, and post-process time from different implementations.

5.3.2. FPGA Performance

The PL configuration time is 340 ms on average, including the time for loading NN parameters into BlockRAM. The line graph in Figure 10 shows the differences in inference times between CPU, GPU and FPGA. The horizontal and vertical axes present the number of records per test (input buffer size) and the processing time, respectively. By doubling the number of records sent each time, the CPU and GPU inference times increase gradually to approximately 0.92 ms, for both platforms, at the buffer size of $2^{12}$ (4096 records). The CPU inference time rises exponentially to 3.01 ms, while the GPU starts to perform better,

with only 1.44 ms, at the end of the test. HLS and Verilog implementation approaches on the PYNQ-Z2 SoC FPGA return a similar inference time, which starts at approximately 0.44 ms in the beginning and then increases slightly to 0.48 ms at the buffer size of 1024. This time is approximately 1.72 ms when the buffer size is 16,384 ($2^{14}$), which is equal to 9,525,581 records per second.

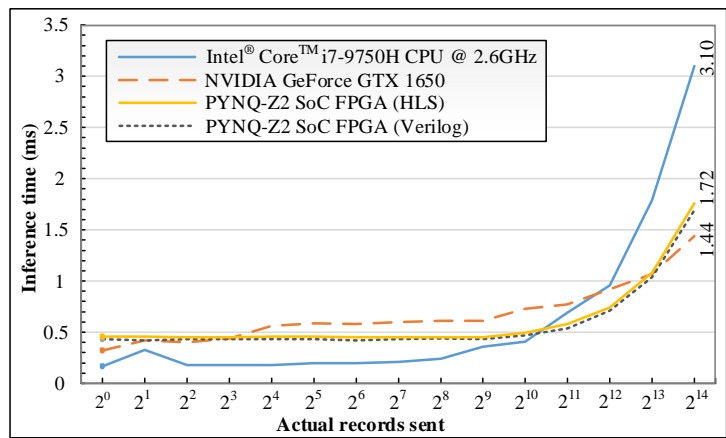

**Figure 10.** The CPU, GPU, and FPGA inference times for different input buffer sizes.

Next, a timer is placed in the inference code after DMA has sent data from the PS to the PL to measure the data transfer time. The timer reported that it takes 0.067 ms, on average, for an input on the PYNQ-Z2 Soc FPGA to be inferred and transferred back to the PS. In addition, the data transfer time from the PS to the PL is close to 0.42 ms. Therefore, the NN block on PYNQ-Z2 SoC, with a Xilinx xc7z020-1clg400c device, is 2.5 and 4.8 times (0.067 ms compared to 0.17 ms and 0.32 ms) faster than Intel® Core™ i7-9750H 2.6 GHz CPU and NVIDIA GeForce GTX 1650 GPU in inference time, respectively. The processing speed of the pipelined design on FPGA can be calculated through the waveform simulation result in Figure 11.

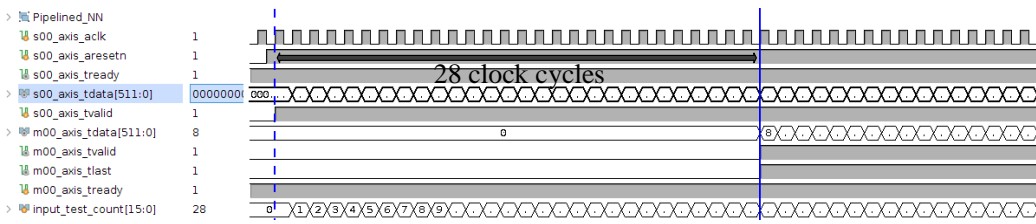

**Figure 11.** The waveform simulation results from the NN on FPGA using a Verilog approach.

The waveform displays the primary input and output AXI4-STREAM interfaces in the NN block. The testbench sends a new input at each clock cycle to demonstrate a network input flow. Each record takes 28 clock cycles to be processed (between the s00_axis_tvalid and m00_axis_tvalid signals). In this pipeline design, the input flow can go through the system continuously without waiting for other processing inputs in the system to be finished; thus, the system overhead for processing one flow is 28 clock cycles, which is equal to 0.28 µs on the PYNQ-Z2 (operating at 100 MHz). This means that the pipelined NN block on the FPGA accelerator at 100 MHz is 53.5 times faster than the max78000EVKIT in simulation. Some factors affected the inference time, in practice, and will be discussed in Section 6.

Finally, the average inference times after ten trials are shown in Table 11. To ensure a fair comparison with the work in [47], the number of records for sending in DMA buffer size was made equal to the number in these works (22,544 records). The authors in [47,50] used the same NSL-KDD dataset for experimenting with their NN models. Our FPGA

implementation is 4.15 times faster than the same platform in [47] and 110.1 times faster than the CPU implementation in [50].

**Table 11.** Comparison in inference time on different platforms and between related works.

| System | Platform | Time (ms) |
|--------|----------|-----------|
| [50] | CPU i3 2.4 GHz | 240.14 |
| [47] | ARM A9 667 MHz | 1458.10 |
| | FPGA accelerator 76 MHz | 9.02 |
| HH-NIDS | **FPGA accelerator 100 MHz (HLS)** | **2.18** |
| | **FPGA accelerator 100 MHz (Verilog)** | **2.24** |

## 6. Discussion

The HH-NIDS framework has trained lightweight NN models on the UNSW-NB15 and IoT-23 datasets using different schemes aimed at two hardware acceleration platforms. Although the eleven inputs used in this prototype are static features, the trained models have achieved high accuracy compared to the related works. To improve the results (higher accuracy and lower the false alarm rate), the Data Pre-processing block in the HH-NIDS framework needs to be upgraded to extract a more comprehensive input feature set.

In Section 5.3.1, CPU and GPU implementations have an advantage in pre-process and post-process times thanks to high operating frequencies and rich memories for buffering input data; the data transfer buses in these platforms are also optimised for a broader range of applications. The GPU also has extra overhead for loading data into its on-chip memory, which explains why it starts to perform better with a greater input buffer size setup.

The MAX78000 microcontroller with NN accelerate processors performs well, requiring only 15 μs to infer an input. The power consumption is ultra-low at 18 mW, while it draws from 15 W to 17 W on the NVIDIA GeForce GTX 1650 GPU to infer inputs. Besides, the NN models on FPGA have shown high overhead in data transferring between the PS and the PL; the actual inference time was explained in the waveform simulation results in Section 5.3.2.

While the HLS implementation is more flexible in changing the NN architecture, the Verilog implementation with pipelining is optimised for the best processing performance. Even though the pipelined design on the PL has no bottle-neck points that possibly operate at the on-chip frequency with a fixed 28 clock cycles overhead, the pre-process and post-process blocks on the PYNQ-Z2 SoC FPGA are run on the PS (ARM-based) with limited speed.

## 7. Conclusions

This paper presents the HH-NIDS framework for a heterogeneous hardware-based implementation of anomaly detection in IoT networks. The framework includes dataset pre-processing, model generation and hardware-based inference on hardware accelerators. The proposed framework is tested with the UNSW-NB15 and IoT-23 datasets using NN models and achieves the highest accuracy of 99.66%. The inference phase is implemented using different approaches on the microcontroller and SoC FPGA. Firstly, the inference phase is implemented on the MAX780000EVKIT, which is 11.3 and 21.3 times faster than Intel® Core™ i7-9750H 2.6GHz CPU and NVIDIA GeForce GTX 1650 GPU, respectively; the power drawn was between 17mW and 18mW when the NN model is inferring. In addition, the inference phase on SoC FPGA is implemented using HLS and Verilog approaches, achieving the same practical processing speed. However, the pipelined design on the PYNQ-Z2 board with the Xilinx Zynq xc7z020-1clg400c device has been optimised to run at the on-chip frequency with a fixed 28 clock cycles overhead. The simulation results have reported a speedup of 53.5 times compared to the MAX78000EVKIT. Future work will

consider using FPGA network specialised hardware (e.g., NetFPGA-SUME) for studying high-performance intrusion detection systems.

**Author Contributions:** Methodology, D.-M.N., A.T., C.P.-Q., N.-T.T. and C.C.M.; Software, D.-M.N.; Validation, D.-M.N. and D.L.; Formal analysis, D.-M.N., C.C.M. and E.P.; Writing—original draft, D.-M.N.; Writing—review & editing, A.T., C.C.M. and E.P.; Supervision, E.P.; Project administration, E.P. All authors have read and agreed to the published version of the manuscript.

**Funding:** This research is supported in part by a grant from Science Foundation Ireland INSIGHT Centre for Data Analytics (Grant number 12/RC/2289-P2) which is co-funded under the European Regional Development Fund.

**Data Availability Statement:** The data presented in this study are available on request from the corresponding author.

**Acknowledgments:** The authors acknowledge the University College Cork (UCC) and Ho Chi Minh City University of Technology (HCMUT), VNU-HCM for supporting this study. We would like also to acknowledge the support from Qualcomm, Analog Devices, AMD/Xilinx and Dell for various parts of this project.

**Conflicts of Interest:** The authors declare no conflict of interest.

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
