# Peer review of "HH-NIDS: Heterogeneous Hardware-Based Network Intrusion Detection Framework for IoT Security"

_futureinternet, doi:10.3390/fi15010009_

Round 1

Reviewer 1 Report

As a general note, the paper is in many cases too technical, "leaning" towards a white paper. Regardless of its scientic value, some implementation details should have been ommited. In other words, it gives me the feeling of reading the Readme file in GitHub.
For example, paragraphs 4.4.1, 4.4.2 do not really offer any scientific value.

I would suggest to eliminate some acronyms and details from the Abstract as for standard guidelines of IEEE.
Such details (e.g., CPU, NVIDIA, etc.) belong to the main body of the paper.

Latest IoT Intrusion detection papers and softwarized implementations are missing,  especially comparative analyses papers to support the main argument against signature-based IDSs.

Paragraphs in lines 48-61 is mixing the IoT device limitations and issues, with GPUs and FPGAs. It needs to be rewritten, clearly distinguishing what goes where, and how/why it is used.

The related work is good and comprehensive, although in order for the reader to grasp a coherent view, I would prefer one or two tables with the basic scores, the hardware used, and the implementation type

Attacks in Tables 1, and 2 need (possibly in another table) to be explained, analyzed as for type, severity, etc.

Maybe, a flowchart for paragraphs 226-237 would be helpful

As a final remark, I would like to see the paper at least two-three pages shorter, with implementation details omitted.

Errata:
IPv4 not Ipv4
uss uses

Reviewer 2 Report

The authors proposed a hardware-based network intrusion detection system for IoT networks.

The authors have not included names and affiliations in the manuscript.

The technical content, the flow of information and the organization of the paper are good.

Author Response

On behalf of all authors of the paper, I would like to thank you for your precious feedback and comments on the content of our manuscript. We took all comments into account very carefully and modified the manuscript accordingly.

Point 1: The authors have not included names and affiliations in the manuscript

Response 1: The authors' names and affiliations will be provided in the next iteration of the manuscript submission.

With the changes we made, we hope that we addressed all the reviewer’s questions and suggestions. Thank you for your consideration of our manuscript and for providing valuable comments which improved the manuscript.

Yours sincerely,

The Authors

Reviewer 3 Report

The work is done properly, and the experiment is very interesting and relevant. The gap is that the process is finite after the readings are done. If we are talking about IDS, then such a system works continuously, and the number of attacks is not uniform. Therefore, there may be a discrepancy in energy consumption. For example, if we have a DDoS attack, which the IDS must constantly recognize, it is logical that the energy load will be large in terms of time. In the future, it is recommended to explain why such hardware should be chosen when performing such experiments. Also, an experiment should be carried out in time stretching, which would be a realistic observation of IDS energy. 

Such an experiment did not reduce the value of the work at all, and it will be published.

Author Response

On behalf of all authors of the paper, I would like to thank you for your precious feedback and comments on the content of our manuscript. We took all comments into account very carefully and modified the manuscript accordingly.

Point 1: There is no analysis as to why this particular hardware should be chosen.

Response 1: At the beginning, we aimed at FPGA for implementing a customised security engine (ANN). The PYNQ-Z2 SoC FPGA and MAX78000 were chosen due to the availability of the equipment in our research lab, with an emphasis on lower cost and versatility. Similarly, the MAX78000 is an ultra-low power Convolutional Neural Network (CNN) inference engine to run Artificial Intelligence (AI) that was introduced in April 2021. The related works in using this MAX78000 are very limited. 

Point 2:  A comparison of energy consumption over time is very difficult. And

what if this kind of attack happens all the time and requires constant training? 

Response 2: We agree that "A comparison of energy consumption over time is very difficult".

Firstly, the training phase is conducted on CPU and GPU platforms, which are high power consumption.

Secondly, the trained models are lightweight and suitable for hardware accelerators (FPGA and Microcontroller). The hardware used for inference in the HH-NIDS framework is more efficient in energy consumption compared to GPUs.

Thirdly, the power consumption of the PYNQ-Z2 and MAX78000EVKIT are reported in the paper. We applied the approach named Offline learning, which means the dataset is ready and labelled. The training should only be repeated when new training strategies are applied.

In conclusion, thanks for addressing "what if this kind of attack happens all the time and requires constant training". This is what we are aiming at in the next version of our framework that uses an Online learning approach (or continuous learning, federated learning).

With the changes we made, we hope that we addressed all the reviewer’s questions and suggestions. Thank you for your consideration of our manuscript and for providing valuable comments which improved the manuscript.

Yours sincerely,

The Authors